# Fibroblast Activation Protein-Targeting Minibody-IRDye700DX for Ablation of the Cancer-Associated Fibroblast with Photodynamic Therapy

**DOI:** 10.3390/cells12101420

**Published:** 2023-05-18

**Authors:** Esther M. M. Smeets, Daphne N. Dorst, Gerben M. Franssen, Merijn S. van Essen, Cathelijne Frielink, Martijn W. J. Stommel, Marija Trajkovic-Arsic, Phyllis F. Cheung, Jens T. Siveke, Ian Wilson, Alessandro Mascioni, Erik H. J. G. Aarntzen, Sanne A. M. van Lith

**Affiliations:** 1Department of Medical Imaging, Radboud University Medical Center, 6525 GA Nijmegen, The Netherlands; esther.markus-smeets@radboudumc.nl (E.M.M.S.);; 2Department of Experimental Rheumatology, Radboud Institute for Molecular Life Sciences, 6525 GA Nijmegen, The Netherlands; 3Department of Surgery, Radboud University Medical Center, 6525 GA Nijmegen, The Netherlands; 4Bridge Institute of Experimental Tumour Therapy, West German Cancer Centre, University Hospital Essen, University of Duisburg-Essen, 47057 Essen, Germany; 5Division of Solid Tumour Translational Oncology, German Cancer Consortium (DKTK Partner Site Essen) and German Cancer Research Centre, DKFZ, 69120 Heidelberg, Germany; 6ImaginAb, Inglewood, CA 90301, USA

**Keywords:** minibody, targeted photodynamic therapy (tPDT), cancer-associated fibroblast (CAF), fibroblast activation protein (FAP), pancreatic ductal adenocarcinoma (PDAC)

## Abstract

Fibroblast activation protein (FAP), expressed on cancer-associated fibroblasts, is a target for diagnosis and therapy in multiple tumour types. Strategies to systemically deplete FAP-expressing cells show efficacy; however, these induce toxicities, as FAP-expressing cells are found in normal tissues. FAP-targeted photodynamic therapy offers a solution, as it acts only locally and upon activation. Here, a FAP-binding minibody was conjugated to the chelator diethylenetriaminepentaacetic acid (DTPA) and the photosensitizer IRDye700DX (DTPA-700DX-MB). DTPA-700DX-MB showed efficient binding to FAP-overexpressing 3T3 murine fibroblasts (3T3-FAP) and induced the protein’s dose-dependent cytotoxicity upon light exposure. Biodistribution of DTPA-700DX-MB in mice carrying either subcutaneous or orthotopic tumours of murine pancreatic ductal adenocarcinoma cells (PDAC299) showed maximal tumour uptake of ^111^In-labelled DTPA-700DX-MB at 24 h post injection. Co-injection with an excess DTPA-700DX-MB reduced uptake, and autoradiography correlated with FAP expression in the stromal tumour region. Finally, in vivo therapeutic efficacy was determined in two simultaneous subcutaneous PDAC299 tumours; only one was treated with 690 nm light. Upregulation of an apoptosis marker was only observed in the treated tumours. In conclusion, DTPA-700DX-MB binds to FAP-expressing cells and targets PDAC299 tumours in mice with good signal-to-background ratios. Furthermore, the induced apoptosis indicates the feasibility of targeted depletion of FAP-expressing cells with photodynamic therapy.

## 1. Introduction

Fibroblast activation protein (FAP) is a type II integral serine protease that is expressed by activated fibroblasts. It facilitates remodelling of the extracellular matrix and therefore is involved in functions such as cell adhesion, invasion, and motility. In over 90% of epithelial tumours, FAP expression is found on the cancer-associated fibroblasts (CAFs). These cells have a critical role in promoting tumour growth, invasion, metastasis, and immunosuppression and ultimately in the clinical outcome of patients [1,2,3,4,5]. The development and application of tracers for non-invasive imaging FAP-expressing cells have therefore gained widespread interest over the past years [6,7].

Besides imaging, targeted depletion of FAP-expressing cells is an appealing treatment option. Depletion of FAP-expressing cells was investigated in transgenic mice with FAP-promotor driven diphtheria toxin receptor (DTR) [5] or using pharmacological FAP inhibitors [8], FAP-targeting vaccines and T-cell therapies [9,10,11,12], and FAP-targeting molecules coupled to cytotoxic compounds [13,14,15,16,17]. These compounds inhibit tumour growth and induce anti-tumour immune responses in various preclinical models; however, some studies also describe severe systemic toxicity, including cachexia and anaemia [18,19]. This indicates the importance of FAP-expressing cells in maintaining normal muscle mass and haematopoiesis. Furthermore, it warrants caution, as FAP expression is also found in wound healing and normal tissues such as the placenta, uterine stroma, embryonic tissue, and multipotent bone marrow stromal cells [5,20,21].

Targeted photodynamic therapy offers a potential solution to these limitations. Here, a targeting compound conjugated to a photosensitizer is activated by light of a specific wavelength, inducing production of reactive oxygen species and resulting in cell death. As opposed to non-targeted photodynamic therapy using the photosensitizer alone, in this approach, off-target toxicity is avoided. Furthermore, as the photosensitizer conjugate is only activated in the region of interest, the effects of eliminating target-expressing cells in other tissues or organs are avoided. The silica-phthalocyanine derivative IRDye700DX is highly suitable and often used for targeted photodynamic therapy, as the water-soluble axial ligands increase hydrophilicity, and the NHS ester enables conjugation to targeting proteins such as antibodies. Furthermore, it is activated with near-infrared light (690 nm), is very photostable, and = has a high singlet oxygen quantum yield [22,23,24].

We and others have proven the feasibility of FAP-targeted photodynamic therapy with a monoclonal antibody [25,26,27,28,29,30,31] or with nanoparticles [32,33]. Though high binding affinity and long circulation time often lead to high uptake of monoclonal antibodies at the target site, smaller molecules could lead to better tissue penetration. In addition, faster blood clearance could lead to a shorter time interval between injection and exposure to light. Other studies have shown similar or increased efficiency of photodynamic therapy with smaller molecules such as nanobodies (~15 kDa), diabodies (~45 kDa), or minibodies (~75 kDa) when compared to a monoclonal antibody [34,35]. Here, we optimized FAP-targeted photodynamic therapy with a minibody. These are comprised of antigen binding single-chain fragment variable regions fused to the C_H_3 region of the human Fc domain. Minibodies maintain the specificity and avidity of full-length antibodies while being biologically inert and lacking immune effector functions. The use of minibodies in non-invasive imaging has shown successful in preclinical [36,37] and initial clinical results [38,39].

Here, we generated a minibody conjugated to the phthalocyanine-based photosensitizer IRDye700DX for application in photodynamic therapy and the chelator DTPA for ^111^In-labeling and quantification or visualization of the tracer in biodistribution and imaging studies. We aim to provide proof of concept for the feasibility of FAP-targeted photodynamic therapy in FAP-expressing cell lines in vitro and in a syngeneic murine model carrying subcutaneous and orthotopic tumours of a murine pancreatic ductal adenocarcinoma (PDAC) cell line, as PDAC is a tumour type in which FAP-expressing CAFs play a large role.

## 2. Materials and Methods

### 2.1. Minibody Conjugation and Characterization

A minibody directed at binding human FAP was developed by ImaginAb Inc. (Inglewood, CA, USA). The minibody was conjugated to p-isothiocyanatobenzyl DTPA (ITC-DTPA, Macrocyclics, Plano, TX, USA) and N-Hydroxysuccinimide IRDye700DX (NHS-IRDye700DX, Licor, Lincoln, NE, USA). NHS-IRDye700DX was added in a 5-fold molar excess in 10% *v*/*v* NaHCO_3_, pH 8.5, and incubated for 1 h, and then, ITC-DTPA was added in a 15-fold molar excess in 10% *v*/*v* NaHCO_3_, pH 9.5, and incubated for 1 h at room temperature. Conjugates were subsequently dialyzed in PBS (pH 7.4) with 1 g/L Chelex (Bio-Rad laboratories Inc., Hercules, CA, USA) to remove unconjugated probes. UV–visible absorbance spectra of 5 µM NHS-IRDye700DX and DTPA-700DX-MB were recorded at 300–800 nm with the Tecan Infinite 200 Pro (Tecan, Mannedörf, Switzerland). Emission spectra at 650–800 nm were recorded after 620 nm excitation.

### 2.2. Radiolabeling with ^111^In and Quality Control

DTPA-700DX-MB was incubated with 0.4 MBq/µg [^111^In]InCl_3_ (Curium, Petten, The Netherlands) for in vitro studies and with 0.042 MBq/µg [^111^In]InCl_3_ for in vivo studies and twice the volume of 0.5 M 2-(N-morpholino)ethanesulfonic (MES) buffer, pH 5.5, for 30 min at room temperature. Labelling efficiency and radiochemical purity were determined by instant thin-layer chromatography (ITLC) on a silica-gel chromatography strip (Agilent Technologies, Amstelveen, The Netherlands), using 0.1 M citrate buffer, pH 6.0, as the mobile phase. Furthermore, we determined the purity of the non-conjugated minibody, the conjugate DTPA-700DX-MB, and 0.1 MBq/µg ^111^In-labelled DTPA-700DX-MB with fast protein liquid chromatography (FPLC), using an Agilent Technologies 1260 Infinity with Yarra 3 µm SEC 3000 column (300 × 7.8 mm, Phenomonex, Torrance, CA, USA) with 0.1 M sodium phosphate 10% 2-propanol as eluent at a flow rate of 0.7 mL/min.

### 2.3. Cell Culture

PDAC299 cells (derived from a pancreatic tumour of a Ptf1a^WT/Cre^;Kras^WT/LSL-G12D^;P53^LSL-R172H^/^fl^ mouse [40]) were cultured in DMEM with glutamax (Gibco) supplemented with sodium-pyruvate and 10% FCS at 37 °C in a humidified atmosphere with 5% CO_2_. NIH-3T3 fibroblasts and NIH-3T3 fibroblasts stably transfected with murine FAP (3T3-FAP; PETR4906, developed by Roche) were cultured in DMEM with glutamax, supplemented with 10% FCS, penicillin, and streptomycin (and 1.5 µg/mL puromycin in case of 3T3-FAP) at 37 °C in a humidified atmosphere with 5% CO_2_.

### 2.4. In Vitro Binding and Internalization of DTPA-700DX-MB

3T3, 3T3-FAP, and PDAC299 cells were grown to 80% confluency in 6-well plates and incubated with 1600 Bq ^111^In-labelled DTPA-700DX-MB in binding buffer (DMEM with 0.5% BSA) at 37 °C for 1, 2.5, or 24 h. After washing twice with PBS, cells were incubated with ice-cold 0.1 M acetic acid and 154 mM NaCl (pH 2.6) for 10 min at 4 °C to collect the membrane-bound fraction. After washing, cells were lysed using 0.1 M NaOH, the lysate was collected, and activity in both fractions was counted in a γ-counter (2480 Wizard 3″, LKB/Wallace, Perkin-Elmer, Boston, MA, USA).

### 2.5. IC50 Determination

3T3-FAP cells were grown to 90% confluency in 12-well plates. Cells were incubated with increasing concentrations of unlabelled DTPA-700DX-MB (0.015 to 100 nM) in the binding buffer, in the presence of 1600 Bq of ^111^In-labelled DTPA-700DX-MB for 4 h on ice. Subsequently, cells were washed twice with PBS and lysed using 1 mL 0.1 M NaOH. Activity in the lysates was counted in a γ-counter (2480 Wizard 3′’, LKB/Wallace, Perkin-Elmer, Boston, MA, USA). The IC50 values were calculated in Graphpad Prism version 5.0.

### 2.6. Singlet Oxygen Production

250 nM DTPA-700DX-MB was incubated with 50 µM p-nitrosodimethylaniline (RNO) and 400 µM imidazole as an acceptor of singlet oxygen in PBS pH 7.4, in clear, flat-bottom 96-well plates (Costar) and exposed to 690 nm light at 200 mW/cm^2^ with a light emitting diode (LEDfactory, Leeuwarden, The Netherlands) [41]. Absorbance at 440 nm was measured every minute up to 13 min (156 J/cm^2^ total light dose) with the Tecan Infinite^®^ 200 Pro (Tecan, Männedorf, Switzerland) to determine singlet oxygen-induced bleaching of RNO.

### 2.7. In Vitro Targeted Photodynamic Therapy with DTPA-700DX-MB

3T3, 3T3-FAP, and PDAC299 cells were grown to 80% confluency in 96-well plates and incubated with DTPA-700DX-MB in binding buffer (DMEM with 0.5% *w*/*v* BSA, BB) or with BB alone at 37 °C in a humidified atmosphere with 5% CO_2_ for 2.5 h. After washing with PBS, regular culture medium was added. The cells were then exposed to 0 or 60 J/cm^2^ 690 nm light at a fluency rate of 200 mW/cm^2^. Cell viability was measured 24 h after tPDT using the CellTiter-Glo assay (Promega, Madison, WI, USA). Furthermore, 3T3-FAP cells were labelled with Vybrant DiO dye according to manufacturer’s protocol (Thermo Fisher Scientific, Waltham, MA, USA) and cocultured with PDAC299 tumour cells in Costar 96-well flat-bottom, clear plates in DMEM with 10% FCS. Cells were incubated with 3 nM DTPA-700DX-MB for 2 h at 37 °C, and upon washing, the plate was irradiated with 60 J/cm^2^ 690 nm light. After 24 h, the cells were incubated with 1 µg/mL propidium iodide (PI) in PBS. After washing twice with PBS, cells were imaged with the EVOS FL cell imaging system with the suitable LED cubes (PI = RFP; DiO = GFP).

### 2.8. Animals

The Radboud University animal ethics committee approved study protocols (CCD number AVD1030020209645). All procedures were performed according to the Institute of Laboratory Animal Research guide for Laboratory Animals. All 6–8-week-old female C57BL/6 mice (6JRj, Janvier) were acclimatized for 1 week upon arrival at our institution. They were fed standard chow ad libitum and housed on a 12 h day–night cycle.

Subcutaneous model—Mice were injected subcutaneously at one or both shoulders with 5 × 10^5^ PDAC299 tumour cells resuspended in 100 µL of DMEM. Mice were weighed, and tumours were measured with a calliper twice a week. Tumour size was calculated as follows: tumour size = (4/3) * π * (L/2) * (W/2) * (D/2), with L being tumour length, W being tumour width, and D being tumour depth. When (one of the) tumours reached a size of 200 mm^3^, usually within 10–14 days, animals were included in the experiments.

Orthotopic model—The pancreas was exposed by making a surgical incision just below the spleen. Either 5 × 10^3^ (*n* = 5) or 2.5 × 10^4^ (*n* = 4) PDAC tumour cells resuspended in 30 µL of DMEM and Matrigel (1:1) were injected into the pancreas with a 27G needle. The wound was closed with a surgical suture, and mice were weighed and imaged with ultrasound (Vevo 2100 preclinical imaging system, Visual Sonics, Amsterdam, The Netherlands, MS-550D transducer) twice per week. At 32 days post injection, when the tumours reached a size of 300–500 mm^3^, mice were included in the experiment.

### 2.9. Biodistribution of ^111^In-Labelled DTPA-700DX-MB

Subcutaneous model—0.3 nmol 10 MBq (for autoradiography analyses and ex vivo biodistribution) or 1 MBq (for ex vivo biodistribution only) ^111^In-labelled DTPA-700DX-MB in 200 µL PBS was injected intravenously in mice carrying PDAC299 subcutaneous tumours. At 4 h (*n* = 5), 24 h (*n* = 7), and 48 h (*n* = 5) post injection, mice were sacrificed through CO_2_/O_2_ asphyxiation. Three additional mice were co-injected with a 7.4-fold excess non-conjugated and unlabelled minibody and sacrificed at 24 h to determine specificity of uptake. Tumour, blood, and organs were collected, weighed, and counted in a γ-counter (WIZARD, 2480 Automatic Gamma Counter, Perkin-Elmer, Boston, MA, USA). Uptake in the tissues was calculated as the percentage of the injected activity per gram of tissue (%IA/g). Tumours of the mice injected with 10 MBq were processed for autoradiography analyses as described below.

Orthotopic model—0.3 nmol 10 MBq (for autoradiography analyses and ex vivo biodistribution) or 1 MBq (for ex vivo biodistribution only) ^111^In-labelled DTPA-700DX-MB in 200 µL PBS was injected intravenously in mice carrying orthotopic PDAC299 tumours (5 × 10^3^ cells *n* = 3; 2.5 × 10^4^ cells *n* = 4). Two additional mice in the group injected with 5 × 10^3^ tumour cells were co-injected with a 7-fold excess non-conjugated and unlabelled minibody. At 24 h post injection, mice were sacrificed through CO_2_/O_2_ asphyxiation. Tumour, blood, and organs were collected, weighed, and counted in a γ-counter (WIZARD, 2480 Automatic Gamma Counter, Perkin-Elmer, Boston, MA, USA). Uptake in the tissues was calculated as the percentage of the injected activity per gram of tissue (%IA/g). Tumours of the mice injected with 10 MBq were processed for autoradiography analyses as described below.

### 2.10. MicroSPECT/CT

A SPECT/CT scan was acquired for 30 min 24 h post injection of one mouse carrying an orthotopic PDAC299 tumour with 10 MBq ^111^In-labelled DTPA-700DX-MB using a U-SPECT/CT-6 (MILabs, Utrecht, The Netherlands). Images were acquired using a 1 mm diameter pinhole ultra-high-sensitivity mouse collimator. The SPECT scan was reconstructed with three iterations and a voxel size of 0.4 mm (MILabs reconstruction software version 12.00), and the image was made with VivoQuant software (Version 2020R2 build9).

### 2.11. Autoradiography

Tumours were harvested and formalin-fixed and paraffin-embedded (FFPE). They were cut into 4 µm sections and mounted on superfrost glass slides. A phosphor screen (Fuji Film BAS-IP SR 2025, Raytest, Straubenhardt, Germany) was exposed to the slides for 72 h in a Fujifilm BAS cassette 2025. Then, the images were acquired with a Typhoon FLA 7000 laser scanner (GE healthcare Life Sciences, Chicago, IL, USA) at a pixel size of 25 × 25 µm. Images were analysed with Aida Image analysis software (Version 4.21, Elysia Raytest, Germany).

### 2.12. In Vivo FAP-tPDT with DTPA-700DX-MB in the Subcutaneous PDAC299 Model

Mice carrying subcutaneous tumours of PDAC299 on both shoulders were injected with 0.6 nmol DTPA-700DX-MB in PBS (*n* = 6) or PBS alone (*n* = 5). At 24 h post injection, mice were anaesthetized with isoflurane (1.5% in 1 L/min oxygen). Uptake in both tumours was visualized with fluorescence imaging (Excitation filter 640 nm, Emission filter Cy5.5 (IVIS Lumina II, PerkinElmer, Waltham, MA, USA)), and then, one of the tumours was exposed to 50–200 J/cm^2^ 690 nm light at 230–300 mW/cm^2^ (DTPA-700DX-MB group: one dose of 55 J/cm^2^ at 280 mW/cm^2^ (*n* = 1), three doses of 55 J/cm^2^ at 280 mW/cm^2^ (*n* = 1), two doses of 100 J/cm^2^ at 230 mW/cm^2^ (*n* = 1), and one dose of 100 J/cm^2^ at 230 mW/cm^2^ (*n* = 3); and the PBS group: one dose of 100 J/cm^2^ at 230 mW/cm^2^ (*n* = 5)). We irradiated the left tumour unless indicated otherwise. The rest of the body was shielded from light using paper towels and aluminium foil. After irradiation, the treated tumour was imaged for fluorescence again. Mice were sacrificed at 24 h post light irradiation with CO_2_/O_2_ asphyxiation. Tumours, livers, kidneys, and spleens were obtained and formalin-fixed and paraffin-embedded.

### 2.13. Histology and Immunohistochemistry

FFPE tumours were sectioned at 4 µm thickness. HE and Sirius red staining were performed using standard protocol. Immunohistochemistry (IHC) was performed with rabbit-anti-FAP (EPR20021, ab 207178, Abcam, Cambridge, UK, 1:100 in PBS/1%BSA, binds to both human and murine FAP) and rabbit-anti-cleaved caspase-3 (9661S, Cell signalling technologies, 1:4000 in PBS/1%BSA) antibodies. First, the slides were deparaffinized and rehydrated.

For the FAP staining, antigen retrieval was performed by heating in EDTA for 10 min at 96 °C in a PT Module (Thermo Fisher Scientific, Waltham, MA, USA). The endogenous peroxidase activity was blocked by incubating with 3% H_2_O_2_ in PBS for 10 min at room temperature, and endogenous biotin was blocked with a biotin/avidin blocking kit (VECTASTAIN, Thermo fisher) for 15 min at room temperature each. Non-specific binding was blocked through preincubation with 20% normal goat serum for 30 min at room temperature. The slides were subsequently incubated with the primary antibody for 1 h at room temperature and with secondary biotinylated goat-anti-rabbit (VECTASTAIN, 1/200 in PBS 1%/BSA) for 30 min, followed by labelling with the avidin–biotin complex (VECTASTAIN, Thermo Fisher, 1/100). The antibody complex was visualized using 8 min incubation with diaminobenzene (bright DAB, Immunologic, VWR, Dublin, Ireland). All slides were counterstained with haematoxylin (Klinipath, Olen, Belgium) for 5 s and mounted with a cover slip (Permount, Thermo-Fisher, Waltham, MA, USA).

For the cleaved caspase-3 staining, antigen retrieval was performed by heating in 10 mM citrate, pH 6.0, for 10 min at 96 °C in a PT Module (Thermo Fisher Scientific, Waltham, MA, USA). The peroxidase activity was blocked by incubating with 3% H_2_O_2_ in PBS. Non-specific binding was blocked through preincubation with 20% normal goat serum for 30 min at room temperature. The slides were subsequently incubated overnight at 4 °C with the primary antibody and with secondary biotinylated goat-anti-rabbit (VECTASTAIN, Thermo Fisher, 1/200 in PBS 1%/BSA) for 30 min, followed by labelling with the avidin-biotin complex (VECTASTAIN, Thermo Fisher, 1/100). The antibody complex was visualized using 8 min incubation with bright DAB. All slides were counterstained with haematoxylin for 5 s and mounted with a Permount cover slip.

### 2.14. Automated Quantification Cleaved Caspase-3 IHC

Slide digitization was performed using a 3DHistech P1000 scanner digital slide scanner (3DHistech, Budapest, Hungary) with a 20× objective at a resolution of 0.24 µm/pixel. To quantify the amount of cleaved caspase-3 after FAP-tPDT, the total tumour area was manually annotated on tissue sections from two different depths in the tumour (approximately 150 µm apart). Tissue folds and staining artefacts were excluded from the annotation. A previously developed automated colour deconvolution algorithm [42] was used to extract the DAB staining from the background haematoxylin staining. This algorithm automatically determines the optimal stain matrix per slide, which is used for unmixing the slide for an optimal colour unmixing. To achieve this, an adapted version of the deconvolution of Ruifrok et al. was used [43]. We extended this algorithm by computing the ratio (positive pixels for a staining per region of interest) using automated Otsu thresholding on a resolution of 2 µm/pixel, which resulted in a thresholded image. The ratio was calculated on this thresholded image by dividing the stain of interest by the total amount of stained pixels.

### 2.15. Distribution Visualisation FAP IHC

To compare the autoradiography signal with the FAP IHC staining, we used the same algorithm to analyse the FAP staining. Instead of using the ratio as output, the thresholded image was used. To compare the FAP IHC staining with autoradiography, which are different in terms of resolution, we generated FAP IHC density maps, which mimic the resolution of the autoradiography (25 μm/pixel). A density map was calculated based on the thresholded binary image of the FAP staining. For this purpose, we defined a circular kernel of 0.49 mm^2^ at a resolution of 32 μm/pixel and applied it to the slide in a sliding window fashion, computing the ratio of positive pixels over the circle area for every position in the slide. To limit the analysis to valid pixels in the tumour area within the slide, only positions within the tumour annotations were considered while sliding the circular kernel.

### 2.16. Statistics

All quantitative data are expressed as mean ± SD. The statistical analyses were performed using GraphPad Prism (Version 5.0, GraphPad Software Inc., San Diego, CA, USA). Student’s *t*-test was used to determine significance.

## 3. Results

### 3.1. DTPA-700DX-MB Binds to FAP-Expressing Cells and Causes Light-Induced Toxicity

The FAP-binding minibody was conjugated to DTPA-ITC and IRDye700DX-NHS using standardized protocols, reaching a ratio of 1.25 IRDye700DX molecules per minibody (Figure 1A). Though FPLC showed an increase of 5.93% to 14.88% of aggregates upon conjugation (Appendix A), spectral properties of the IRDye700DX were retained upon conjugation (Figure 1B). The construct was labelled with ^111^In in 0.5 M MES buffer (pH 5.4) for 20 min, and successful labelling was determined with ITLC and FPLC (Appendix A). Membrane binding of DTPA-700DX-MB to 3T3-FAP cells was confirmed (3.4 ± 0.07, 6.7 ± 0.9, and 11.1 ± 0.13 percentage of added activity at 1, 2.5, and 24 h incubation, respectively), and the construct was efficiently internalized (6.8 ± 0.14, 11.8 ± 0.41, and 16.2 ± 0.98 percentage of added activity at 1, 2.5, and 24 h incubation, respectively) (Figure 1C). Lack of binding or internalization in native 3T3 cells as well as in 3T3-FAP cells pre-incubated with an excess unlabelled minibody confirm FAP specificity. The half-maximal inhibitory concentration of DTPA-700DX-MB was 44 nM (Figure 1D). The conjugate produced singlet oxygen upon irradiation with 690 nm light with similar efficiency as the non-conjugated NHS-IRDye700DX (Figure 1E). Upon incubation with varying doses of DTPA-700DX-MB and subsequent irradiation with 60 J/cm^2^ 200 mW/cm^2^ 690 nm light, efficient cell death was induced (Figure 1F). This effect was not observed when incubating native 3T3 cells or without irradiation, indicating receptor specificity and lack of dark toxicity of DTPA-700DX-MB (Figure 1F). Since on some tumour cell lines expression of FAP is observed, we investigated the binding of DTPA-700DX-MB to PDAC299 cells, but no binding or light-induced cytotoxicity was observed (Appendix A). This indicates that these cells do not express FAP, which was also verified in co-cultures of PDAC299 and 3T3-FAP, in which only 3T3-FAP cells were killed upon treatment with DTPA-700DX-MB and light (Appendix A).

### 3.2. DTPA-700DX-MB Targets Subcutaneous PDAC299 Tumours In Vivo

PDAC299 is a tumour cell line that is derived from the pancreatic tumour of a *Ptf1a^WT^*^/*Cre*^*;Kras^WT^*^/*LSL-G12D*^
*P53^LSL-R172H^*^/*fl*^ mouse [40]. Upon subcutaneous injection into C57BL/6 mice, these tumours show a moderately well-differentiated morphology organized in glandular structures that are typical for adenocarcinomas (Figure 2A, HE). Furthermore, FAP-expressing fibroblasts were present (Figure 2A, FAP IHC), and excessive collagen formation was found in these tumours (Figure 2A, Sirius red). In a time-optimization biodistribution in mice carrying subcutaneous tumours of PDAC299, a maximum relative tumour uptake of 7.3 ± 1.02 %IA/g and a tumour-to-blood ratio of 8.3 ± 0.82 were observed at 24 h post injection of 0.3 nmol ^111^In-labelled DTPA-700DX-MB (Figure 2B,C and Appendix A). Tumour uptake was significantly lower upon co-injection of a 7.4x excess of unlabelled and unconjugated minibody (4.4 ± 1.45 %IA/g, *p* = 0.0059), indicating at least a partly FAP-specific uptake (Figure 2B and visualization of IRDye700DX-derived fluorescence in Figure 2D). The main clearance route of DTPA-700DX-MB is the liver, and some uptake was observed in the kidney and spleen. Furthermore, FAP-dependent uptake was observed in the tibia (9.85 ± 1.09 vs. 3.64 ± 0.80 %IA/g upon blocking, *p* < 0.0001), which is most likely due to presence of FAP-expressing cells in the bone marrow. Autoradiography of PDAC299 tumour sections showed heterogenous distribution of DTPA-700DX-MB in the tumour (Figure 2E). To compare the autoradiography signal with the FAP IHC staining, which are different in terms of resolution, we generated FAP IHC density maps, which mimic the resolution of the autoradiography. In general, regions with a high signal on autoradiography also showed a higher signal on FAP IHC density maps visually, while in regions with a low signal on autoradiography, little FAP expression was observed on immunohistochemistry (Figure 2E and Appendix A).

### 3.3. DTPA-700DX-MB Induces Cell Death in Subcutaneous PDAC299 Tumours In Vivo

To determine in vivo efficacy, mice carrying two PDAC299 tumours were injected with 0.6 nmol DTPA-700DX-MB (*n* = 6) or PBS as a control (*n* = 5). One of the tumours was exposed to 690 nm light. Bleaching of IRDye700DX-derived fluorescence indicated efficient activation of the photosensitizer. A light dose of 55 J/cm^2^ at a dose rate of 280 mW/cm^2^ was not sufficient to induce apoptosis (*n* = 1,Appendix A). Irradiation with three separate doses of 55 J/cm^2^ at 280 mW/cm^2^ and two separate doses of 100 J/cm^2^ at 230 mW/cm^2^ light did induce apoptosis; however, in these conditions, we saw immediate bleeding of the skin and induction of crust formation, most probably due to excessive heating (both *n* = 1,Appendix A). A single dose of 100 J/cm^2^ at 230 mW/cm^2^ did not result in bleeding or crust formation but did induce apoptosis, as illustrated by the increase in caspase-3-positive fraction when compared to the contralateral non-irradiated tumours (*n* = 3, 0.41 ± 0.17 vs. 0.01 ± 0.01) (Figure 3A and Appendix A). In the group injected with PBS, some upregulation of apoptosis was observed as well when comparing the caspase-3-positive fraction of the irradiated tumour with the non-irradiated tumour (*n* = 5, 0.23 ± 0.15 vs. 0.029 ± 0.012); however, this effect was less apparent than for the minibody-treated group (*p* = 0.0432 when comparing the MB and PBS light-exposed groups, Figure 3B and Appendix A).

### 3.4. DTPA-700DX-MB Targets PDAC299 Orthotopic Tumours In Vivo

Maximum relative tumour uptakes of 8.97 ± 2.01 %IA/g and 9.28 ± 1.44 %IA/g were observed at 24 h post injection of 0.3 nmol ^111^In-labelled DTPA-700DX-MB in mice injected with 5 × 10^3^ and 2.5 × 10^4^ tumour cells, respectively (Figure 4A and Appendix A). Tumour uptake was 7.89 ± 2.03 %IA/g upon co-injection of a 7-fold excess of unlabelled and unconjugated minibody (not significant). As in the mice carrying subcutaneous tumours, we observed uptake in the liver, kidney, and spleen. Furthermore, FAP-dependent uptake was observed in the tibia (6.61 ± 0.55 vs. 4.15 ± 0.12 %IA/g upon blocking, *p* = 0.0096). The signal in the orthotopic tumour could be visualized with SPECT/CT imaging (Figure 4B, tumour area is encircled). Autoradiography of PDAC299 tumour sections showed heterogenous distribution of DTPA-700DX-MB. Similar to the subcutaneous tumours, comparable patterns were observed for the autoradiography and FAP-IHC density maps in the orthotopic tumours (Figure 4C and Appendix A).

## 4. Discussion

FAP-expressing CAFs are shown to have pro-tumourigenic effects in multiple solid tumour types. In this work, we characterize DTPA-700DX-MB, which can be used for light-induced ablation of FAP-expressing cells.

The anti-FAP minibody was efficiently functionalized with the photosensitizer IRDye700DX, which is one of the most suitable photosensitizers for targeted photodynamic therapy due to its high hydrophilicity, photostability, and singlet oxygen quantum yield [22]. We demonstrated efficient and specific binding to murine FAP-transfected 3T3 fibroblasts and toxicity upon exposure to 690 nm light. In vivo, we showed targeting to both subcutaneous and orthotopic PDAC299 tumours. The uptake is at least partly specific, as shown by inhibition of accumulation after administration of an excess unlabelled minibody and colocalization of the minibody with FAP expression, as determined immunohistochemically. The limited molar excess of minibody (7-fold as compared to up to 100-fold used in other blocking studies) could lead to an underestimation of FAP-specific uptake. In the in vivo therapy experiment, we observed induction of apoptosis in the PDAC299 subcutaneous tumours, as illustrated by cleaved caspase-3 expression, in both the minibody and PBS-treated groups following light exposure. The upregulation in the minibody-treated group was, however, more pronounced, and part of the upregulation of caspase-3 expression may have been induced by excessive heating [44]. Surprisingly, we also observed a low auto-fluorescent signal in the tumours of PBS-injected mice, which was bleached upon irradiation. Though the origin of this fluorescent signal is unknown and has not been described in other studies, this may also be another contributor to the increase of caspase-3 staining that we observed in these tumours.

In these studies, we used a syngeneic mouse model PDAC299 using a murine cell line with PDAC typical KRAS and Trp53 mutations. We have shown that the subcutaneous PDAC299 tumours have an abundant stroma with collagen bundles containing FAP-expressing fibroblasts. As this model shows physiological overexpression of FAP on the stromal component, as is seen in human tumours as well, we believe that it better recapitulates the physiological situation as opposed to models that have been used in other studies, such as mixtures of cancer cells and fibroblasts and human tumour cell lines with either physiological or artificial FAP overexpression. Though the subcutaneous model is adequate to provide insight into pharmacokinetics and targeting properties of the DTPA-700DX-MB, we do believe that further studies into the biological effects of the FAP-targeted PDT should be investigated in orthotopic models because those better reflect the local environment in which the tumours develop.

Other groups have employed FAP-targeted PDT with a nanoparticle and an antibody [26,30,31,32,33]. Zhen et al. showed efficient killing of fibroblasts in the murine breast cancer model 4T1, which resulted in increased CD8+ T-cell infiltration. Katsube et al. reported efficient killing of CAFs in a murine model that was co-inoculated with human squamous carcinoma cells (TE4) and activated human foetal oesophageal fibroblasts (FEF3), which improved the sensitivity to conventional chemotherapy. These results indicate the potential therapeutic efficacy of FAP-targeted photodynamic therapy in multiple tumour types. As mentioned above, further studies in an orthotopic PDAC299 model with varying light and protein doses are needed to determine the effect on immunomodulation and efficacy of other systemic therapies such as chemo- and immunotherapies.

Besides FAP-targeted PDT, radionuclide therapy with FAP-targeting small molecules has gained attention over the past years. FAP-targeting small molecules labelled with actinium-226 or lutetium-177 showed modest growth inhibition in preclinical studies, and initial clinical studies are emerging [45,46,47]. Though systemic toxicities are a potential problem of radionuclide therapies, an advantage is that the therapeutic effect occurs in all lesions in the body, and it is thus suitable for application in metastasized disease. PDT can only be used for tumours that are accessible to light that has a limited penetration depth (up to 1 cm for NIR light). We therefore envision clinical application of FAP-tPDT to be applied intraoperatively or through multiple laser fibres for larger and more deeper-seated lesions of localized disease. Furthermore, we expect this therapy to be applied in an adjuvant manner, sensitizing the tumours for other systemic therapies such as chemotherapy and immunotherapy.

## 5. Conclusions

In conclusion, we have shown that the DTPA-700DX-MB conjugate efficiently binds to FAP-expressing cells and induces cell death upon light exposure in vitro. Furthermore, it targets FAP-expressing fibroblasts in PDAC299 subcutaneous and orthotopic tumours and induces apoptosis in the subcutaneous tumours upon light exposure. This study provides proof-of-concept of FAP-targeted PDT in PDAC, and future studies should demonstrate the feasibility in other tumour types and the effects on tumour growth, immunomodulation, and efficacy of other systemic therapies.

## Figures and Tables

**Figure 1 cells-12-01420-f001:**
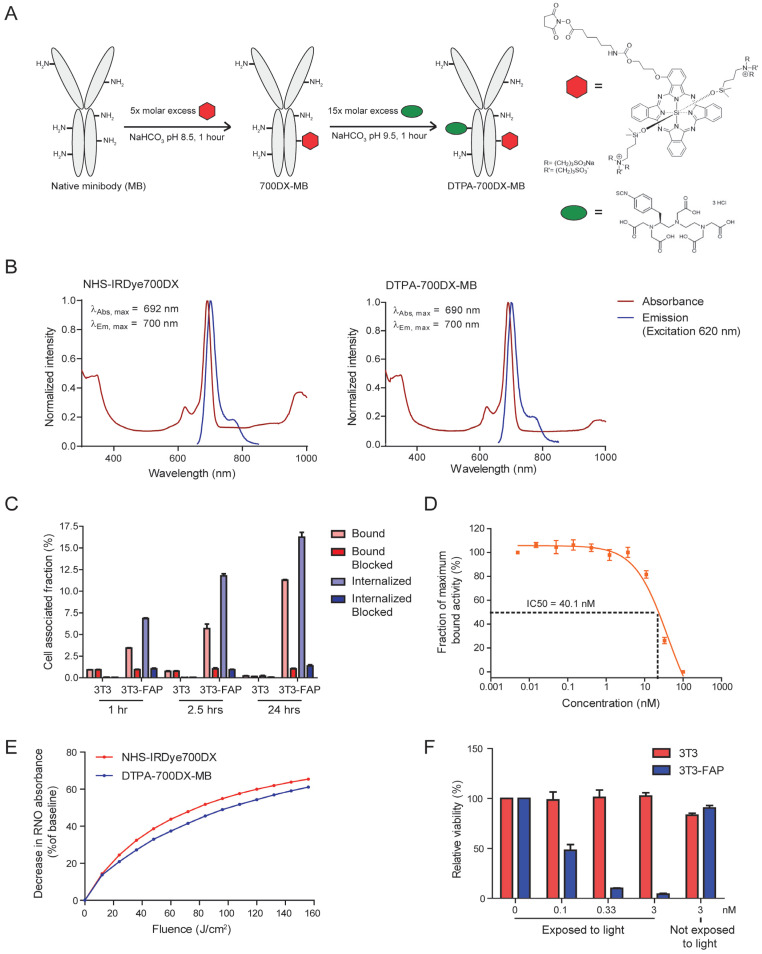
In vitro characterization of DTPA-700DX-MB. (**A**) Scheme for conjugation of IRDye700DX-NHS and DTPA-ITC to the minibody to generate DTPA-700DX-MB. (**B**) Normalized photophysical spectra (absorbance and emission at 620 excitation) of 5 µM NHS-IRDye700DX (left) and 5 µM DTPA-700DX-MB (right) in PBS. (**C**) Bound and internalized fractions of ^111^In-labelled DTPA-700DX-MB after 1, 2.5, or 24 h incubation of 3T3 or 3T3-FAP cells at 37 °C. (**D**) Half-maximal inhibitory concentration. (**E**) Singlet oxygen production as measured by bleaching of reporter molecule p-nitrosodimethylaniline (RNO). (**F**) Cell viability of 3T3 and 3T3-FAP cells after incubation of varying doses of DTPA-700DX-MB for 2.5 h and subsequent irradiation with 60 J/cm^2^ 200 mW/cm^2^ 690 nm light. Non-irradiated samples are taken as controls to determine dark toxicity of DTPA-700DX-MB.

**Figure 2 cells-12-01420-f002:**
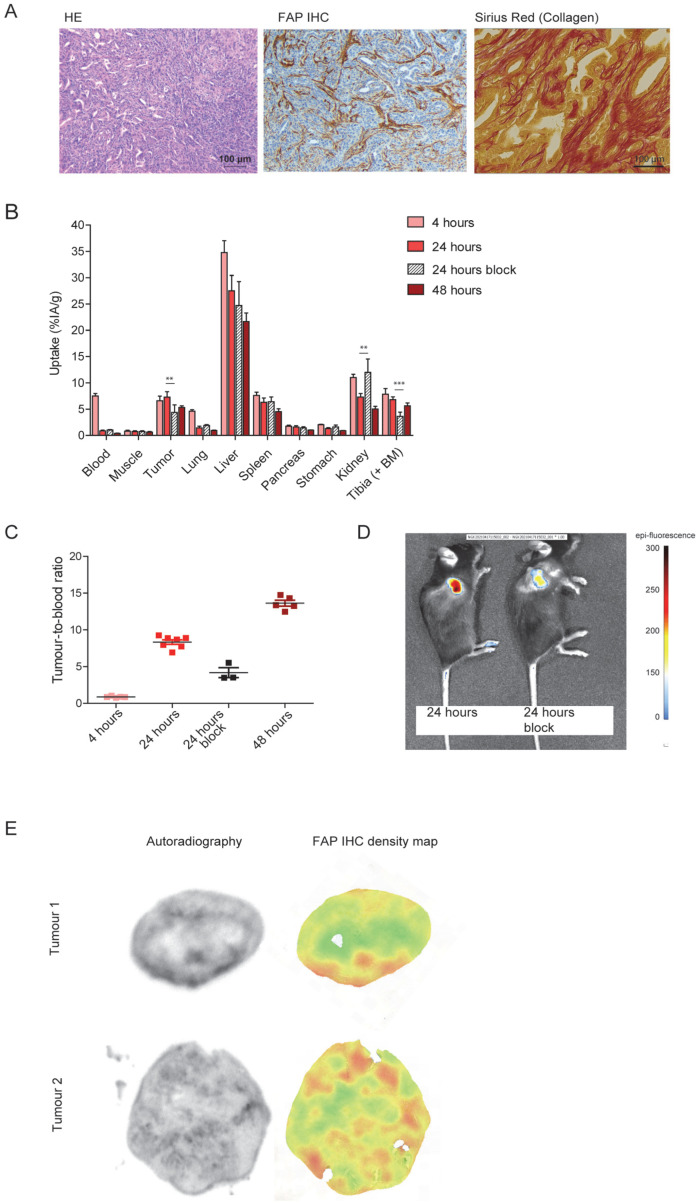
In vivo tumour targeting of DTPA-700DX-MB in subcutaneous PDAC299. (**A**) Characterization of the PDAC299 model, showing excessive tumour stroma formation and presence of activated fibroblasts, as illustrated by FAP expression and collagen deposition, as shown in the Sirius red staining. (**B**) In vivo biodistribution depicted as percentage of injected activity dose per gram of tissue at 4, 24, or 48 h after injection of 0.3 nmol 1 MBq ^111^In-labelled DTPA-700DX-MB. One group of mice was co-injected with a 7.4-fold excess unlabelled minibody. ** *p* < 0.01, *** *p* < 0.001. (**C**) Tumour-to-blood ratios calculated from the biodistribution data. (**D**) Fluorescence imaging of two mice at 24 h post injection of 0.3 nmol ^111^In-labelled DTPA-700DX-MB; one mouse was co-injected with a 7.4-fold excess unlabelled minibody (right). (**E**) Autoradiography of PDAC299 tumour sections upon injection of 0.3 nmol 10 MBq ^111^In-labelled DTPA-700DX-MB and FAP IHC density maps, which represent the FAP IHC staining in resolution comparable to the autoradiography.

**Figure 3 cells-12-01420-f003:**
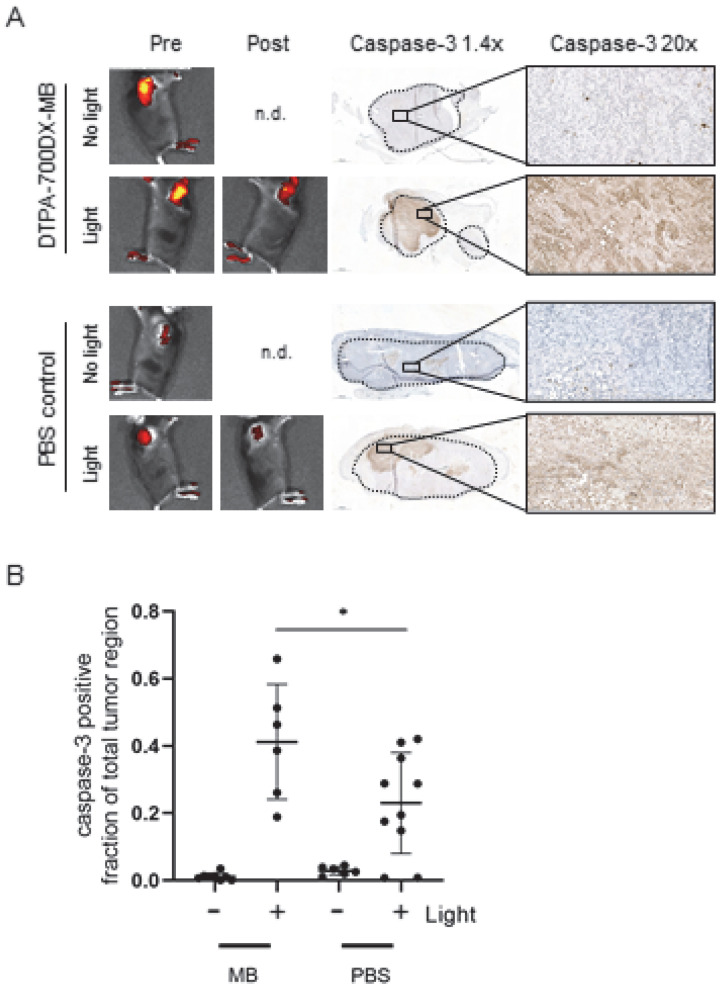
In vivo efficacy of targeted photodynamic therapy with DTPA-700DX-MB in subcutaneous PDAC299. (**A**) Pictures are representative images from the minibody-treated and the PBS control groups. Mice carrying subcutaneous PDAC299 tumours were injected with 0.6 nmol DTPA-700DX-MB 24 h after injection fluorescence was visualized (pre), then tumours were exposed to 100 J/cm^2^ 230 mW/cm^2^ of 690 nm light, and fluorescence was visualized again (post). Tumours were formalin-fixed and paraffin-embedded, and induction of apoptosis was assessed with IHC. (**B**) Automated quantification of the percentage caspase-3-positive region of the whole tumour region, for two sections per tumour. * *p* < 0.05.

**Figure 4 cells-12-01420-f004:**
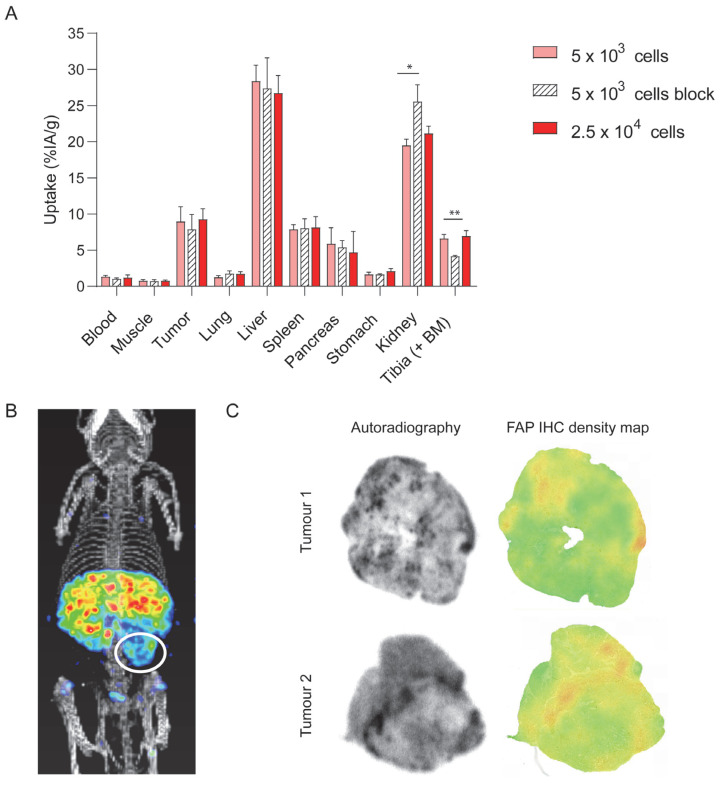
In vivo tumour targeting of DTPA-700DX-MB in an orthotopic model. (**A**) In vivo biodistribution depicted as percentage of injected activity dose per gram of tissue at 24 h after injection of 0.3 nmol 10 MBq ^111^In-labelled DTPA-700DX-MB in mice carrying orthotopic PDAC299 tumours. Two mice were co-injected with a 7-fold excess unlabelled minibody. * *p* < 0.05, ** *p* < 0.01. (**B**) Micro-SPECT/CT imaging of a mouse at 24 h post injection of 0.3 nmol 10 MBq ^111^In-labelled DTPA-700DX-MB. (**C**) Autoradiography of orthotopic PDAC299 tumour sections upon injection of 0.3 nmol 10 MBq ^111^In-labelled DTPA-700DX-MB and FAP IHC density maps, which represent the FAP IHC staining in resolution comparable to the autoradiography.

## Data Availability

The datasets used and/or analysed during the current study are available from the corresponding author on reasonable request.

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
