# Peer review of "Fibroblast Activation Protein-Targeting Minibody-IRDye700DX for Ablation of the Cancer-Associated Fibroblast with Photodynamic Therapy"

_cells, 2023, doi:10.3390/cells12101420_

Round 1

Reviewer 1 Report

The expression of fibroblast activation protein (FAP) is strongly correlated with cancer-associated fibroblasts (CAFs). In this manuscript, a FAP-binding minibody was conjugated to the chelator diethylenetriaminepentaacetic acid (DTPA) and the photosensitizer IRDye700DX (DTPA-700DX-MB). Biodistribution of DTPA-700DX-MB and tumour uptake of 111In-labeled DTPA-700DX-MB were characterized. The PDT-induced apoptosis indicates feasibility of targeted depletion of FAP-expressing cells. In my opinion, this is an interesting study, but I can't recommend it for publication before addressing the following issues.

1. In the chapter of “3.1. DTPA-700DX-MB binds to FAP-expressing cells and causes light-induced toxicity”, the authors should further characterize the molecular structure to prove that IRDye700DX forms a stable chemical bond with the minibody and DTPA. Please explain the process and mechanism for the preparation of 111In-labelled DTPA-700DX-MB in brief.

2. Please illustrate and label the meanings of red, green and blue in Figure S1.

3. Please clarify the abbreviation in the manuscript and figures, for example, the “Int” in the Figure 1B, the “3x 55 J/cm2” and “2x100 J/cm2” in the Line 370-371. Otherwise, it may cause unnecessary confusion to the reader

4. In the chapter of “3.2. DTPA-700DX-MB targets subcutaneous PDAC299 tumours in vivo” and the Figure 2B, the data units of tumor uptake should be uniform.

5. It is quite confusing that the difference of parallel samples, for example, the “N=6” for the DTPA-700DX-MB, however, the “N=5” for the control. What is even more surprising that, the N=1 for “the light dose of 55 J/cm2 at a dose rate of 280 mW/cm2” and “3x 55 J/cm2 at 280 mW/cm2 and 2x100 J/cm2 at 230 mW/cm2 light”.

Author Response

Thank you for your thoughtful comments and suggestions on our manuscript, they have been helpful, see in attached PDF-file.

Reviewer 2 Report

In this work, the authors report the DTPA-700DX-MB conjugate as a promising photosensitizer for photodynamic therapy of cancer-associated fibroblasts. The binding of the DTPA-700DX-MB conjugate with FAP-expressing cells and its PDT effects upon light exposure in vitro and in vivo have been investigated, showing considerable results. However, this work still requires critical improvements to be suitable for publication in Cells.

1. In the introduction part, the author should briefly describe photodynamic therapy and its limitations with current photosensitizers.

2. The revised manuscript should include the reason for choosing phthalocyanine as the photosensitizer core.

3. The authors should provide the synthetic scheme for the DTPA-700DX-MB conjugate.

Is it possible to characterize the DTPA-700DX-MB conjugate by 1H NMR?

4. The authors should compare the UV-vis and PL spectra of the starting phthalocyanine and the DTPA-700DX-MB conjugate.

5. The confirmation of ROS generation should be performed using a ROS probe, and live-cell and dead-cell markers should be used to indicate the status of cancer cells.

6. To probe the beneficial effect of the DTPA-700DX-MB conjugate, the authors should compare the PDT effect of the DTPA-700DX-MB conjugate and free phthalocyanine.

Author Response

(The authors gave the same response as above.)

Round 2

Reviewer 2 Report

This work still requires critical improvements to be suitable for publication in Cells.

The authors should reconsider the introduction section and correct all the information. For example, singlet oxygen also belongs to reactive oxygen species, and Phthalocyanine PS is hydrophobic. Suitable references should be provided.

Author Response

Thank you for your valuable comments and feedback on our manuscript.
